# Genomic Insights into *Neofusicoccum laricinum*: The Pathogen Behind Chinese Larch Shoot Blight

**DOI:** 10.3390/jof11050399

**Published:** 2025-05-21

**Authors:** Jialiang Pan, Zhijun Yu, Wenhao Dai, Chunhe Lv, Yifan Chen, Hong Sun, Jie Chen, Junxin Gao

**Affiliations:** 1Center for Biological Disaster Prevention and Control, National Forestry and Grassland Administration, Shenyang 110034, China; panjialiang1987@126.com (J.P.); cebaozx@163.com (Z.Y.); dwh311026@163.com (W.D.); lchhelz@163.com (C.L.); chenyifan960510@163.com (Y.C.); sunhongcaf@163.com (H.S.); 2Key Laboratory of Forest Disaster Warning and Control in Yunnan Province, College of Forestry, Southwest Forestry University, Kunming 650224, China

**Keywords:** *Neofusicoccum laricinum*, larch shoot blight, whole-genome sequencing, adaptation, genetic variation

## Abstract

Larch shoot blight, caused by the fungus *Neofusicoccum laricinum*, threatens larch (*Larix* spp.) forests across northeastern China, jeopardizing both timber productivity and ecological stability. This study aimed to investigate the genomic diversity, population structure, and potential adaptive mechanisms of *N. laricinum* across contrasting climatic regions. To achieve this, we conducted whole-genome resequencing of 23 *N. laricinum* isolates collected from three major provinces—Heilongjiang, Inner Mongolia, and Jilin—that represent distinct climatic zones ranging from cold-temperate to relatively warmer regions. We identified ~219.1 K genetic variants, offering a detailed portrait of the pathogen’s genomic diversity. Population structure analyses, including principal component analysis and phylogenetic tree, revealed clear genetic differentiation aligning with geographic origin and climate. Functional annotation (GO and KEGG) highlighted enrichment in metabolic, stress-response, and membrane transport pathways, suggesting potential adaptation to varied temperature regimes and environmental pressures. Moreover, region-specific variants—particularly missense and stop-gain mutations—were linked to genes involved in ATP binding, oxidoreductase activity, and cell division, underscoring the fungus’s capacity for rapid adaptation. Collectively, these findings fill a critical gap in the population genetics of *N. laricinum* and lay a foundation for future disease management strategies to larch shoot blight under changing climatic conditions.

## 1. Introduction

Larch (*Larix* sp.) is a critical coniferous genus in northern China, renowned for its rapid growth, high-quality timber, and strong environmental adaptability. In forest ecosystems, larch plays an essential role in maintaining biodiversity, serving as a carbon sink, and stabilizing ecological functions. Economically, its timber is extensively used in construction, furniture, and paper industries, making it one of the most important fast-growing timber species [1]. In addition, the tall and graceful form of larch, coupled with seasonal foliage color changes, makes it highly valued in landscape design. However, with the growing expanse of larch plantations, the incidence of larch shoot blight has sharply risen, threatening the full realization of the species’ ecological and economic benefits.

Larch shoot blight is a serious fungal disease caused by *Neofusicoccum laricinum* (*N. laricinum*). This ascomycetous pathogen exhibits high infectivity and a broad host range. Early symptoms typically appear as water-soaked lesions on newly emerged shoots, which spread rapidly and result in shoot wilt. The needles then turn brown from the tip to the base before eventually falling off. In severe cases, tree crowns may die back and side branches can proliferate, forming “witches’ brooms” [2,3]. Under humid conditions, black pycnidia (the conidia-bearing structures of the pathogen) often appear on infected tissues [4]. Optimal infection occurs at 20 °C conditions, and infections principally begin in early August. Notably, wounds are not required for spore penetration, and disease symptoms appear approximately two weeks post-infection. Cool winters and short summers are unfavorable for disease development, and some spores may overwinter within pseudothecia.

*N. laricinum* is primarily distributed in temperate Asia, including eastern China, Japan, the Korean Peninsula, and the Russian Far East [5]. In China, it is classified as a major quarantine disease, mainly affecting larch forests in the Northeast, North, and Southwest regions. The pathogen was first reported in 1938 in Hokkaido, Japan [6], and subsequent studies on its biology and pathogenicity were largely conducted in Japan [7]. Despite its economic importance, genomic research on *N. laricinum* remains limited. However, genomic studies on related *Neofusicoccum* species, such as *N. parvum*, have revealed insights into virulence factors, secondary metabolite biosynthesis, and host–pathogen interactions, providing valuable frameworks for understanding pathogenicity in this genus [8,9]. Recently, the reference genome assembly for *N. laricinum* (ASM2990638v1) has become available in NCBI, indicating a genome organized into 14 chromosomes. Nevertheless, detailed population genomic analyses and studies on genetic diversity, adaptation, and spread patterns of *N. laricinum* are still lacking.

Given the scarcity of genomic and population genetic studies, understanding the genetic structure and diversity of *N. laricinum* is crucial for developing effective management strategies against larch shoot blight. In this study, we collected samples from multiple provinces in China, isolated and identified the pathogen, and used multi-gene SNP data to analyze its genetic diversity and population differentiation. We identified highly polymorphic SNP markers suitable for geographic population analysis and constructed a genetic diversity map to illustrate relationships among distinct populations. These findings provide new insights into the epidemiology and dispersal mechanisms of *N. laricinum*, laying the foundation for region-specific control strategies and the development of a DNA barcode–based detection system for early monitoring and disease management.

## 2. Materials and Methods

### 2.1. Sampling

Samples of the Chinese indigenous *N. laricinum* were collected from various regions across northeastern China in three major provinces—Heilongjiang, Inner Mongolia, and Jilin—spanning different climatic conditions. In the Heilongjiang, samples (*n* = 7) were collected; characterized by a cold-temperate monsoon climate, Heilongjiang experiences long, frigid winters (−20 to −30 °C in the coldest months) and brief, mild summers (~20 to 25 °C). The average annual temperature often hovers around 3 to 5 °C. Sampled sites featured diverse forest landscapes, where larch stands are well-adapted to harsh winter conditions. In the Inner Mongolia, samples (*n* = 3) were collected. The climate in this region is predominantly temperate continental, with large diurnal and seasonal temperature fluctuations. Winters can be severe (average temperatures dropping below −20 °C), while summers are relatively warm (~18 to 24 °C). Rainfall is generally lower than in Heilongjiang or Jilin, and certain areas are semi-arid. Larch forests here endure substantial environmental stresses, including drought and extreme cold. In the Jilin, samples (*n* = 13) were collected (Table 1); Jilin experiences a moderate temperate monsoon climate, with moderately cold winters (~−15 to −20 °C) and warm, humid summers (~22 to 26 °C). Annual average temperatures range between 4 and 9 °C. The region typically receives more summer precipitation than Heilongjiang or Inner Mongolia, promoting relatively lush forest growth “https://en.climate-data.org/asia/china-110/ (accessed on 9 May 2025)”. All 23 *N. laricinum* samples were selected to capture the breadth of ecological and climatic diversity across northeastern China. Each sample thus reflects adaptation to a distinct agro-climatic and ecological context, ranging from cold, arid environments to milder, moisture-rich habitats.

### 2.2. N. laricinum Collection and Isolation

At each sampling site, symptomatic larch trees were selected within plantations, with sampling performed from four cardinal directions (east, south, west, and north). From each selected tree, 20 diseased needles were collected and transported to the laboratory for pathogen isolation.

In the laboratory, single-spore isolation was conducted from lesions on diseased needles following established protocols [10,11]. Needles were surface sterilized by immersion in 75% ethanol for 10 s, followed by treatment with 1% sodium hypochlorite (NaClO) for 40 s, and then rinsed three times with sterile distilled water. The sterilized needles were air-dried on autoclaved filter paper. Under a microscope, individual conidia were picked directly from lesions using fine insect needles and transferred onto potato dextrose agar (PDA) plates. After incubation, single fungal colonies were selected and subcultured for further purification and long-term preservation.

Genomic DNA was extracted from pure cultures of *N. laricinum* isolates collected from different geographic regions using the fungal DNA extraction protocol based on sodium dodecyl sulfate (SDS) [12], conducted at the Shenyang Institute of Technology and Southwest Forestry University.

### 2.3. Genomic DNA Quality Control, Library Preparation, and Sequencing

Genomic DNA concentration was quantified using the Qubit^®^ 2.0 fluorometer (Life Technologies, CA, USA) with the dsDNA BR assay kit (Life Technologies, Carlsbad, CA, USA). DNA quality was assessed by 1% agarose gel electrophoresis to detect degradation and RNA contamination. DNA purity was further evaluated by measuring the OD260/OD280 ratio using a Nanodrop spectrophotometer (Thermo Fisher Scientific, Waltham, MA, USA), and only samples with concentrations ≥ 300 ng/μL were retained for library preparation.

Next-generation sequencing libraries were prepared using the ND627 kit (Vazyme Biotech Co., Ltd., Nanjing, China), following the manufacturer’s instructions. The prepared libraries were sequenced on the DNBSEQ-T7 platform (MGI Tech Co., Ltd., Shenzhen, China) in paired-end mode with a read length of 150 bp, employing the DNBSEQ-T7 High-throughput Sequencing Set (MGI Tech Co., Ltd., Shenzhen, China) for sequencing chemistry.

### 2.4. Short Read Pre-Processing, Variants Calling, and Quality Control

Raw sequencing data underwent initial quality assessment using FastQC (v0.11.8) [13]. Reads with an N content exceeding 10%, a Phred quality score below 20, or a length shorter than 20 bp were removed. Adapter trimming was subsequently performed using Trimmomatic (v0.39) [14]. High-quality reads were then aligned to the *N. laricinum* reference genome (assembly ASM2990638v1) using Bwa-mem2 (v2.2.1) [15]. Alignment-produced SAM/BAM files were sorted and index based on genomic coordinates via Samtools (v1.14) [16].

The sorted BAM files served as input for variant calling performed with the Genome Analysis Toolkit (GATK, v4.1.1.0) [17]. Duplicate reads were marked with the MarkDuplicates tool. Single nucleotide polymorphisms (SNPs) and insertions/deletions (InDels) were identified for each sample using GATK HaplotypeCaller in GVCF mode, which performs local de novo haplotype assembly to improve the accuracy of variant detection. The resulting gVCF files from all samples were combined using GenotypeGVCFs for joint genotyping, producing a multi-sample VCF file. To ensure high-confidence variant calls, a series of quality control and filtering steps were applied using Bcftools (v1.9) [18]. Variants located within 5 bp of an InDel and InDels occurring within 10 bp of each other were removed. Clustered variants (defined as more than two variants within a 5 bp window) were also excluded. Additional filtering criteria included a Phred-scaled quality score (QUAL) ≥ 30, a quality-by-depth (QD) value ≥ 2.0, a mapping quality (MQ) ≥ 40, and a Fisher strand bias (FS) ≤ 60.0. Default filtering recommendations from GATK were also followed. The resulting high-confidence variant set was used for downstream genetic and functional analyses. Variant density across the genome was visualized using the R package ggplot2 v 3.5.1 [19].

### 2.5. Population Structure Analysis with Principal Component Analysis and Phylogenetic Tree

Population structure among isolates was assessed using principal component analysis (PCA) conducted with PLINK (v2.0) [20]. Additionally, a genetic distance matrix was calculated using PLINK v2.0 [19] to infer phylogenetic relationships among individual isolates. The resulting Neighbor-Joining (NJ) phylogenetic tree was constructed based on this matrix and visualized using the R package APE v5.8-1 [21]. ADMIXTURE v1.3.0 [22] was used to quantify genome-wide admixture increasing *K* from 1 to 3. The ADMIXTURE plot was visualized using the R package ggplot2 v 3.5.1 [19].

### 2.6. Annotation Analysis and Functional Enrichment Analyses

Variants were annotated using SnpEff v5.0 [23]. Functional annotation of the identified variants was conducted by aligning gene sequences, including both novel and previously annotated genes, against several public databases using NCBI BLAST version 2.9.0 [24] with an *E*-value threshold of 1 × 10^−5^. The databases used for annotation included eggNOG/COG [25], GO [26], KEGG [27], NCBI non-redundant protein database (Nr) [28], Pfam [29], and Swiss-Prot [30]. Functional enrichment analysis was performed using the R package clusterProfiler v 4.9.1 [31], applying the enrichGO and enrichKEGG functions to identify significantly enriched GO terms and KEGG pathways. Pathways with a *p*-value less than 0.05 were considered significantly enriched.

## 3. Results

### 3.1. Phenotypic Observations of Larch Shoot Blight

Larch shoot blight (caused by *N. laricinum*) is a destructive fungal disease primarily infecting the newly emerged, non-lignified shoots of larch (*Larix* sp.). Early symptoms include chlorosis and darkening of the shoot tips, resin exudation, and needle wilt, eventually leading to branch dieback and, in severe cases, top-kill of seedlings.

Shortly after infection, the tender stems or axes of new larch shoots lose their green color and gradually turn from light brown to dark brown or almost black. A slight shrinkage or narrowing is often visible, and resin frequently oozes from infected areas. The upper portion of the shoot typically curves downward like a hook, and most needles wither and drop off—often leaving just a small tuft of needles at the tip (Figure 1a). By the following spring, large black fruiting bodies (*pseudothecia*) and smaller pycnidia appear in bark crevices at infected shoot tips, leading to dieback of all tissues above the infection site. In seedlings, infection can kill the main leader, resulting in a “headless” seedling (Figure 1b).

When collected branches were examined, those displaying severely curved, hook-like shoot tips were selected for closer inspection. Upon observing the apex of these drooping shoots, numerous dark fruiting bodies were found. Larger black dots represented the fungus’s ascostromata, whereas smaller black points were pycnidia, confirming the pathogenic characteristics typical of larch shoot blight (Figure 1c).

### 3.2. Distribution of Variants in Chinese Indigenous N. laricinum Isolates

A total of 23 Chinese indigenous *N. laricinum* isolates were collected from diverse geographical regions across the provinces of Heilongjiang, Inner Mongolia, and Jilin, which represent cold-temperate and relatively warmer climatic zones. Genomic DNA was extracted from each isolate and subjected to whole-genome resequencing. This process generated approximately 28.1 Gb of short-read data, totaling around 201 million reads, with an average sequencing coverage of approximately 27.6× per isolate (Appendix A). Quality reports revealed that 99.3% of bases had Phred scores above Q20 and 97.49% exceeded Q30. On average, 86.6% of the reads were successfully mapped to the *N. laricinum* reference genome (assembly ASM2990638v1), indicating high data quality and alignment efficiency. Variant calling identified approximately 219,100 sequence variants across the isolates. These included about 208,400 SNPs and 10,700 insertions or deletions. The variants were unevenly distributed across the genome (Figure 2).

### 3.3. Genetic Relationship Among Chinese Indigenous N. laricinum

To elucidate the genetic relationships among Chinese indigenous *N. laricinum*, we analyzed ~208.5 K SNPs across all isolates (Figure 3a). PCA (Figure 3b) showed clear genetic differentiation: PC1 (accounting for ~50% of the variance) primarily separated the Heilongjiang isolates (red squares) from the others, while PC2 (~30% of the variance) further distinguished the Inner Mongolia isolates (blue triangles) from those of Jilin (green circles). The phylogenetic tree (Figure 3c) corroborated these geographic groupings by clustering isolates according to their province of origin. In addition, population structure analysis (Figure 3d) indicated that isolates from each province predominantly group into distinct subpopulations at both *K* = 2 and *K* = 3, highlighting limited but notable admixture events among geographic regions. Overall, these findings suggest that *N. laricinum* populations in China exhibit significant genetic differentiation associated with their respective locations.

### 3.4. Variants Annotation and Enrichment Analyses

To determine the potential functional impacts of the identified variants, we performed a detailed genomic annotation along with gene ontology (GO) and Kyoto Encyclopedia of Genes and Genomes (KEGG) enrichment analyses. Pie charts in Figure 4a summarize the genomic distribution of SNPs and Indels, based on their locations and variant types. For SNPs, nearly half (~45%) were intergenic, while ~23% and ~22% resided in downstream and upstream regions, respectively. A smaller fraction occurred within exons, introns, or splice-site regions. Indels followed a similar pattern, with the majority (~40%) located downstream, ~22% upstream, and ~21.5% intergenic, and relatively few falling into exonic, intronic, or splice-site categories.

The GO enrichment analysis (Figure 4b, Appendix A) revealed that most variants map to genes involved in fundamental biological processes and stress responses (Biological Process category), as well as membrane or organelle-related components (Cellular Component category). In the Molecular Function category, enzymes with binding or catalytic activities were prominent, indicating that the identified variants may affect key metabolic and regulatory functions. KEGG classifications (Figure 4c, Appendix A) further underscored the role of variants in metabolic processes (e.g., energy metabolism, amino acid metabolism) and genetic information pathways (e.g., replication, transcription, translation). Several variants were also associated with membrane transport and environmental adaptation categories. Overall, this highlights how *N. laricinum* has diversified its genomic toolkit to cope with various ecological pressures and potentially associate with its pathogenic capabilities.

### 3.5. Comparison of Core and Region-Specific Variants

To determine how variants are distributed across isolates and whether certain regions harbor unique polymorphisms, we compared the variant profiles of all individuals and identified both common (“core”, Appendix A) and distinct variants (Figure 5a). Each “petal” in the radial diagram represents a single *N. laricinum* isolate, labeled with the number of unique variants found in that sample; the center shows a core set of 82 variants shared by all isolates. Notably, isolates from Jilin—a comparatively warmer region—carried a higher number of unique variants than those from the colder areas (Heilongjiang and Inner Mongolia).

Next, we performed enrichment analyses (GO) on these region-specific variants (Figure 5b, Appendix A). Key Biological Processes involved cellular metabolism and stress responses; in the Cellular Component category, membrane- and organelle-related factors were enriched, and for Molecular Function, catalytic activities and binding capabilities appeared frequently. This aligns with the hypothesis that populations in milder climates may accumulate variants fostering metabolic flexibility or stress tolerance.

To further investigate the impact of gene-coding variants in relation to temperature differences, we extracted missense and stop-gain variants from samples in colder vs. warmer regions and conducted additional GO term analyses (Appendix A). Figure 5c highlights the top ten GO terms by frequency in these gene-coding variants. Among these terms were “ATP binding”, “heme binding”, and “oxidoreductase activity”, reflecting possible shifts in energy metabolism and redox processes that may support fungal survival and pathogenicity under diverse thermal conditions. Enrichment of cellular division and binding-related terms also suggests that these variants could influence growth rate and host interaction. Overall, these results reveal clear genetic signatures distinguishing isolates from cold versus warmer regions, potentially underpinning local adaptation in *N. laricinum* populations.

## 4. Discussion

In this study, we sampled and analyzed whole-genome sequences from 23 indigenous *N. laricinum* isolates collected across diverse geographic zones in three major provinces (Heilongjiang, Inner Mongolia, and Jilin), representing both colder and relatively warmer climates. Then, we then characterized the genetic variants and population structure of 23 Chinese indigenous *N. laricinum*. Last, we annotated these genetic variants through functional gene enrichment analyses, which directly or indirectly linked to environmental adaptation, membrane, metabolic, catabolic, and stress responses. Together, these genetic findings support the hypothesis that the variants observed in the *N. laricinum* genome may have been shaped by adaptation to local ecological conditions.

Worldwide, *N. laricinum* has emerged as a notable pathogen in larch (*Larix* spp.) forests across Asia [32]. In China, its dissemination has been largely attributed to the movement of infected seedlings and plant materials. Although *N. laricinum* typically thrives in warmer, more humid environments, it also exhibits remarkable resilience in colder climates—particularly in Heilongjiang and Inner Mongolia—owing to its broad thermal tolerance [33,34]. Previous work suggests that both genetic and epigenetic mechanisms underpin its ability to endure environmental extremes, including low temperatures [32,34,35]. These observations led us to hypothesize that regional selection pressures may have shaped distinct genetic signatures in *N. laricinum* populations.

Because genetic responses to selective forces often manifest as changes in genetic variants [36,37], we sought to compare *N. laricinum* isolates from contrasting ecosystems—subtropical versus cold-temperate conditions. Our analyses identified approximately 219.1 K genetic variants. Population structure relationships (including PCA and phylogenetic tree) revealed clear genetic differentiation between isolates from colder provinces (Heilongjiang and Inner Mongolia) and those from Jilin, a comparatively warmer region. The formation of province-specific clusters suggests that geographic barriers and local climates may independently shape evolutionary trajectories. In particular, the genetic separation of Heilongjiang and Inner Mongolia isolates may reflect selective pressures associated with colder temperatures and potentially different ecological interactions. Due to limited genetic research on *N. laricinum*, our findings are particularly important. This pattern of environment-driven divergence mirrors observations in other fungal pathogens. For instance, *Cryphonectria parasitica* shows distinct spatial and temporal population structures shaped by historical introductions and local environmental constraints [38,39]. Similarly, *Zymoseptoria tritici* exhibits strong signatures of thermal adaptation and geographic structuring aligned with global wheat cultivation zones [40]. Overall, these findings illuminate how environment-specific adaptations contribute to genetic structuring in *N. laricinum*, reinforcing the view that local ecological factors have played a pivotal role in the evolution and distribution of this pathogen.

Further, by integrating functional annotations from both GO and KEGG pathway analyses, we identified several prominent functional categories likely contributing to the ecological success of *N. laricinum*. GO enrichment highlighted genes associated with stress response, metabolism, and membrane transport, each of which may play a vital role in mitigating temperature fluctuations, managing limited resources, and navigating potential host-defense barriers [41,42]. Notably, variants enriched in GO terms such as “oxidoreductase activity” and “ATP binding” suggest an adaptive shift in energy metabolism, potentially enabling more rapid growth and infection cycles in colder climates [43]. Similar metabolic adaptations have been reported in other fungal pathogens. For instance, in Fusarium graminearum, oxidative stress-response pathways, particularly those involving bZIP transcription factors and glutathione biosynthesis, are critical for environmental resilience and host adaptation [44,45]. Meanwhile, KEGG classifications offered a broader evolutionary context by grouping genes into orthologous sets shared across eukaryotes. Notably, we detected enriched categories for protein turnover, transcription, and signal transduction, suggesting that *N. laricinum* leverages robust regulatory networks to modulate gene expression under stress. Furthermore, genes involved in cell wall, membrane, and envelope biogenesis were over-represented, suggesting structural adaptations that may enhance fungal integrity under cold or desiccating conditions [46]. Such mechanisms parallel those observed in *Botrytis cinerea*, where cold-stress studies have revealed upregulation of cell wall remodeling and protective proteins [47]. Conversely, expansions in KEGG categories associated with carbohydrate transport and metabolism may enable *N. laricinum* to exploit a broader range of carbon sources, supporting ecological flexibility in comparatively colder or nutrient-variable environments [48]. This pattern is also seen in pathogens like *Zymoseptoria tritici* [49] and *Fusarium oxysporum* [50], where diversified sugar transporter and CAZyme repertoires contribute to host adaptation and survival in fluctuating niches.

We explored the distribution of genetic variants within *N. laricinum* isolates collected from distinct environments in northeastern China. Comparative analyses between isolates from cold regions (Heilongjiang and Inner Mongolia) and a warmer region (Jilin) revealed several high-impact, region-specific variants. Notably, isolates from colder regions harbored distinct mutations, including a stop-retained mutation in gene Nla_2G0011680 and a stop-gained mutation in Nla_14G0001920. Both genes are associated with the GO term GO:0009982 (pseudouridine synthase activity), suggesting a possible role in RNA modification and stress adaptation critical for survival in cold environments [51]. These findings are consistent with genomic studies in *Magnaporthe oryzae*, where pseudouridine synthase-related RNA processing genes have been implicated in stress response and pathogenicity under varying environmental conditions [52]. Moreover, GO enrichment analysis revealed functional categories such as oxidoreductase activity, energy metabolism, and heme binding, which point toward potential metabolic adjustments and enhanced redox balance [53]. Such adaptations are aligned with shifts in energy metabolism and oxidative stress responses seen in other fungal pathogens, supporting survival under cold stress [54,55]. Similar mechanisms have also been documented in cold-adapted fungi like *Pseudogymnoascus destructans*, which modulates redox activity and metabolic efficiency for cold tolerance [56].

Collectively, these results demonstrate that *N. laricinum* has undergone genetic adaptations that likely contribute to its ecological flexibility across diverse environments. Our analyses support the hypothesis that *N. laricinum* populations are shaped by localized ecosystem conditions. Given the substantial temperature variability across northeastern China, it appears the pathogen has evolved genomic strategies to persist across a broad thermal gradient. While this adaptability enhances the pathogen’s resilience, it also complicates disease management, especially in forestry systems spanning heterogeneous climates. As a future direction, de novo genome assemblies of *N. laricinum* isolates could provide deeper insights into structural variation, novel gene content, and adaptive evolution beyond what variant-based analyses can capture. By expanding the catalog of adaptive genetic variants, our study lays a foundation for targeted control strategies and underscores how environmental pressures continue to shape pathogen evolution in real time.

## 5. Conclusions

In this study, we analyzed whole-genome sequences of 23 indigenous *N. laricinum* isolates collected from three major provinces in northeastern China (Heilongjiang, Inner Mongolia, and Jilin). By comparing the genomic relationships among these geographically distinct populations, our results demonstrate that local environmental pressures and possible historical factors have influenced the genetic diversity and population structure of *N. laricinum*. Comparative genomic and functional enrichment analyses revealed key adaptive signatures related to stress response, energy metabolism, membrane structure, and RNA modification. These findings offer valuable insights into the ecological adaptability of *N. laricinum* and underscore the importance of considering local adaptation in the development of effective and region-specific disease management strategies.

## Figures and Tables

**Figure 1 jof-11-00399-f001:**
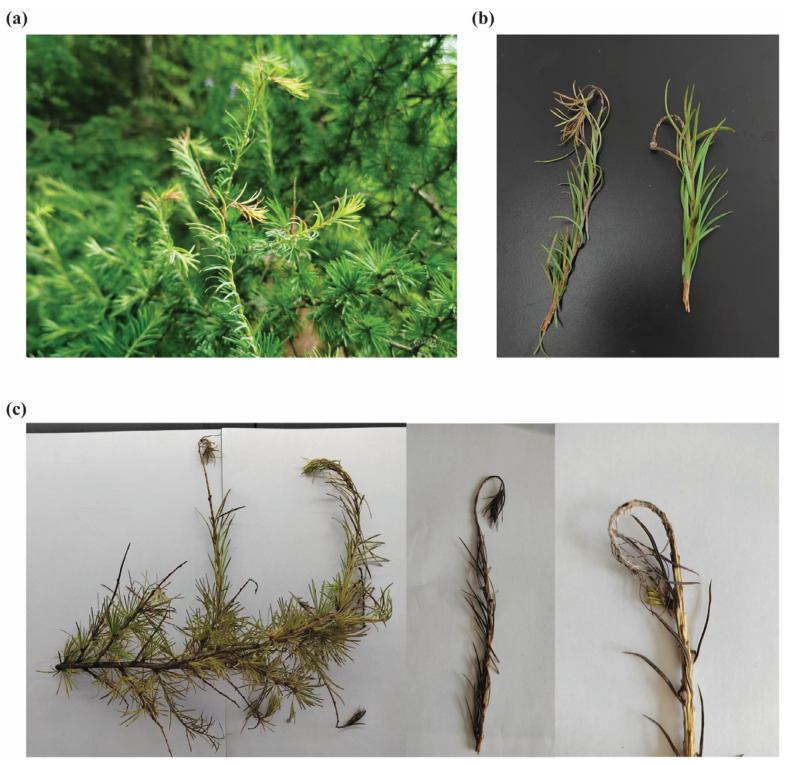
Phenotypic observations of larch shoot blight caused by *N. laricinum.* (**a**) Symptoms at the whole-tree level. (**b**) Symptoms on individual branches. (**c**) Infected larch branches were collected, and top shoots exhibiting a downward, hook-like curvature were selected for closer examination of the pathogen’s fruiting bodies.

**Figure 2 jof-11-00399-f002:**
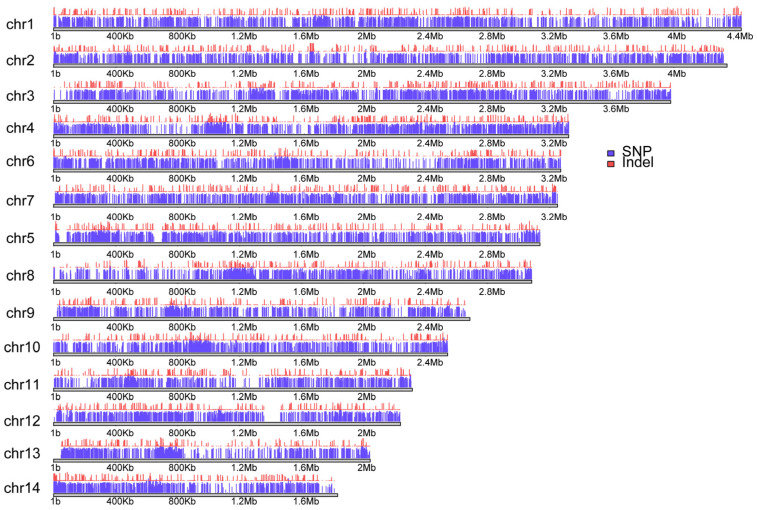
SNP and Indel density distributions in Chinese indigenous *N. laricinum*. The purple plot represents SNP density, and the red plot shows Indel density. The horizontal axis indicates the length of each chromosome.

**Figure 3 jof-11-00399-f003:**
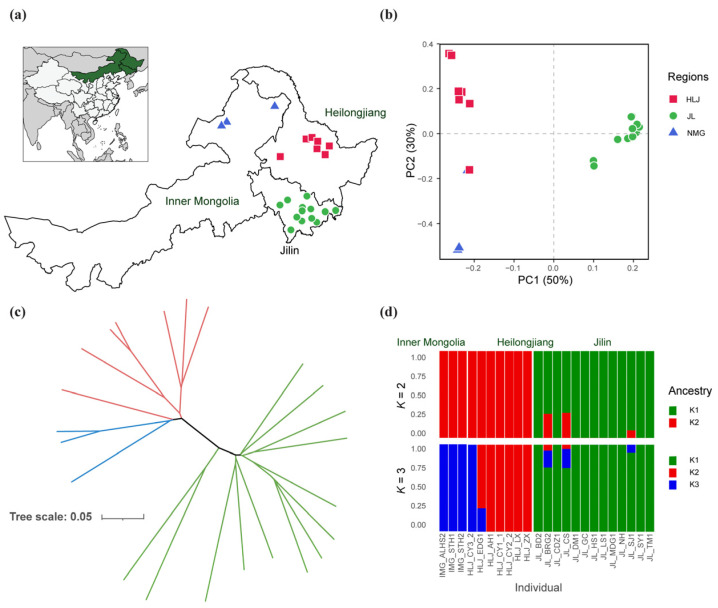
Genetic structure of Chinese indigenous *N. laricinum*. (**a**) Geographic distribution of the sampled isolates, with red squares (Heilongjiang), green circles (Jilin), and blue triangles (Inner Mongolia). The inset highlights these provinces within China. (**b**) Principal component analysis (PCA) of ~208.5 K SNPs. PC1 and PC2 explain 50% and 30% of total variance, respectively, revealing clear regional clustering. (**c**) Unrooted Neighbor-Joining phylogenetic tree based on SNP data. Branch colors correspond to provincial groupings—red for Heilongjiang, green for Jilin, and blue for Inner Mongolia. (**d**) Population structure analysis at *K* = 2 and *K* = 3. Each vertical bar represents a single isolate, with color proportions indicating ancestral components (*K*1, *K*2, *K*3). The x-axis denotes individual sample IDs, grouped by their geographic origin.

**Figure 4 jof-11-00399-f004:**
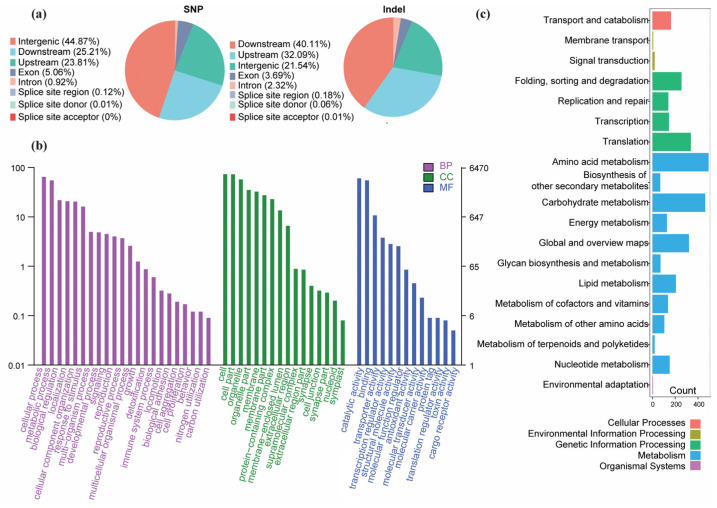
Variant annotation and functional enrichment analyses. (**a**) Pie charts illustrating the proportion of SNPs and Indels across different genomic regions (intergenic, upstream, downstream, exon, intron, and splice-site regions). (**b**) Gene Ontology (GO) term enrichment covering Biological Processes (BPs), Cellular Components (CCs), and Molecular Functions (MFs). The y-axis displays enriched GO terms, and the x-axis denotes enrichment significance on a log scale. (**c**) KEGG enrichment analysis categorizing genes harboring variants into diverse functional groups. Bar lengths represent the count of genes associated with each KEGG category; functional classes are color-coded by overarching pathways.

**Figure 5 jof-11-00399-f005:**
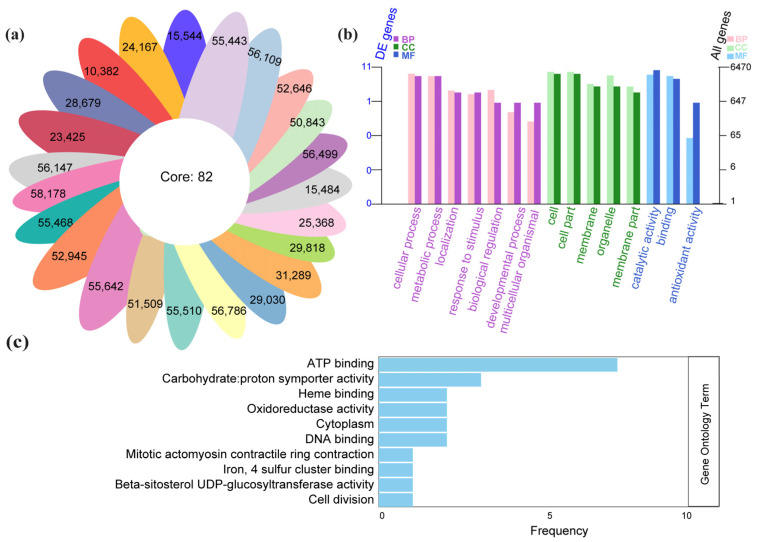
Comparison of core and region-specific variants, and functional enrichment analyses. (**a**) Radial diagram illustrating unique variant counts (petals) for each isolate, plus a central “core” set of 82 variants shared by all samples. Warmer-region isolates (e.g., Jilin) harbor more unique variants overall. (**b**) GO enrichment for region-specific variants, showing terms associated with Biological Processes (BPs), Cellular Components (CCs), and Molecular Functions (MFs). Bars represent the degree of enrichment on a log scale. (**c**) Top ten GO terms among gene-coding variants (missense and stop-gain) identified in comparatively cold vs. warm locales. Key functional categories include ATP binding, oxidoreductase activity, and cell division, suggesting adaptive mechanisms linked to environmental conditions.

**Table 1 jof-11-00399-t001:** Information on samples from indigenous Chinese populations of *N. laricinum* isolates (*n* = 23).

IDs	Species	Host Tree	Country, Provinece	Geographic City, District	No. of Samples
NMG_ALHS2	*Neofusicoccum laricinum*	Larix gmeli-nii	China, Inner Mongolia	Hulun Buir, Alihe	3
NMG_STH1				Hulun Buir, Oroqen Zizhiqi	
NMG_STH2				Hulun Buir, Oroqen Zizhiqi	
JL_GC	*Neofusicoccum laricinum*	Larix gmeli-nii	China, Jilin	Liao Yuan, Dongfeng	13
JL_HS1				Hunchun, Heshang	
JL_TM1				Tumen, Shixian	
JL_SY1				Tumen, Shixian	
JL_BD2				Yanji, Ba Daolin	
JL_DM1				Yanbian, Dongming	
JL_NH				Yanbiang, Wangqing	
JL_BRG2				Yanbiang, Wangqing	
JL_LS1				Yanbian, Helong	
JL_SJ1				Yanbian, Songjiang	
JL_MDG1				Dunhuan, Mu Dangang	
JL_CDZ1				Dunhuan, Qing Dingzi	
JL_CS				Jilin, Changsi	
HLJ_EDG1	*Neofusicoccum laricinum*	Larix gmeli-nii	China, Heilongjiang	Yichun, Yunhao	7
HLJ_CY1_1				Yichun, Yunhao	
HLJ_CY2_2				Yichun, Yunhao	
HLJ_CY3_2				Yichun, Yunhao	
HLJ_LX				Yichun, Yunhao	
HLJ_ZX				Yichun, Yunhao	
HLJ_AH1				Qiqihar, Kedong	

## Data Availability

The raw full-length sequencing data (in FASTQ format) have been submitted to the European Nucleotide Archive (ENA) under the project accession number PRJEB87206 (ERP170428).

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
