# Peer review of "Genomic Insights into Neofusicoccum laricinum: The Pathogen Behind Chinese Larch Shoot Blight"

_jof, 2025, doi:10.3390/jof11050399_

Round 1

Reviewer 1 Report

The proposed article is devoted to the relevant problem of the influence of environmental factors to the population structure of Neofusicoccum laricinum, the causal agent of shoot blight of larch trees grown in China. The Authors have conducted whole-genome resequencing of 23 N. laricinum isolates collected on the host trees Larix gmelinii in three major provinces of China and confirmed the differences between indigenous populations collected in different climatic zones. The genetic diversity of N. laricinum was characterized based on multi-gene SNP data, and highly polymorphic SNP markers suitable for geographic population differentiation were identified. This helped to determine the genetic structure of Chinese indigenous N. laricinum populations. The genetic variants annotated through functional gene enrichment analyses were linked to environmental adaptation, membrane, metabolic, catabolic and stress responses. The results are of special importance since the adaptation of N. laricinum to varying environments under changing climatic conditions can lead to the increased spread of the disease.

1. The Introduction section contains information on the biological features of the causal agent, as well as on the disease harmfulness, occurrences, symptoms and geographical dispersion. The final part of the Introduction section summarizes the main results of the research. However, there is no overview of the relevant genomic studies of the pathogen. Why the current research was necessary? What is the chromosome number in the genome of Neofusicoccum laricinum? The corresponding information should be added in the Introduction section. Introduction should be extended.

2. I consider it is necessary to add additional information to the section” 2. Genomic DNA isolation, sequence library preparation and quality control”, namely:

What method was used for DNA isolation?

Traditionally, pathogen isolates are cultivated on a nutrient medium and the DNA is isolated. Another option allows for the isolation of all DNA types from the affected plants and includes not only DNA of a particular fungal species, but also DNA of the host plant and DNA of other symbionts and pathogens. In this case, it is necessary to specify which programs were used to filter out the target sequences from the total molecular pool.

3. I recommend to extend the Discussion section and consider relevant information on the analogous genomic studies of other fungal pathogens.

4. The Conclusion section should be highlighted at the end of the manuscript.

Formal notes:

5. Line 162 - "3. Genetic relationship among Chinese indigenous Pinewood Nematode". This raises questions, since the article is talking about a pathogenic fungus.

6. Supplementary materials:

The name of Supplementary file 1 is missing.

Supplementary file 4. Please indicate that 23 N. laricinum isolates were studied (the word “isolates” is missing).

7. The figures/tables/images/schemes are appropriate. They properly show the data. All illustrative materials are of satisfactory quality. However, to my opinion, Figures 4 and 5 are too small and will be difficult for a reader to perceive them. 

Author Response

Comments 1: The Introduction section contains information on the biological features of the causal agent, as well as on the disease harmfulness, occurrences, symptoms and geographical dispersion. The final part of the Introduction section summarizes the main results of the research. However, there is no overview of the relevant genomic studies of the pathogen. Why the current research was necessary? What is the chromosome number in the genome of Neofusicoccum laricinum? The corresponding information should be added in the Introduction section. Introduction should be extended.

Response 1: Thank you for your valuable comment. We agree with this comment. Therefore, we have extended the Introduction to include an overview of relevant genomic studies of the pathogen, a clearer explanation of the necessity of this research, and information regarding the chromosome number of Neofusicoccum laricinum. These revisions can be found on page 2, lines 47–68.

Comments 2: I consider it is necessary to add additional information to the section” 2. Genomic DNA isolation, sequence library preparation and quality control”, namely: What method was used for DNA isolation? Traditionally, pathogen isolates are cultivated on a nutrient medium and the DNA is isolated. Another option allows for the isolation of all DNA types from the affected plants and includes not only DNA of a particular fungal species, but also DNA of the host plant and DNA of other symbionts and pathogens. In this case, it is necessary to specify which programs were used to filter out the target sequences from the total molecular pool.

Response 2: Thank you for your valuable comment. We appreciate this important suggestion. In response, we have added detailed information regarding the DNA isolation method used in our study. Specifically, we have separated the content into two sections: "2.2. N. laricinum Collection and Isolation” and “2.3. Genomic DNA Quality Control, Library Preparation, and Sequencing.”

In Section 2.2, we clarify that genomic DNA was extracted from pure cultures of Neofusicoccum laricinum isolates grown on potato dextrose agar (PDA) medium. Single fungal colonies were selected and subcultured to ensure purity, and DNA was extracted using the SDS-based fungal DNA extraction protocol. Therefore, only fungal DNA was obtained, with no contamination from host plant DNA or other microorganisms.

In Section 2.3, we provide detailed descriptions of the methods used for DNA quantification and quality assessment, including the use of the Qubit® 2.0 Fluorometer, 1% agarose gel electrophoresis, and the NanoDrop spectrophotometer. We also describe the next-generation sequencing procedures, including library preparation using the ND627 kit and sequencing on the DNBSEQ-T7 platform (pages 3–4, lines 102–131).

These additions clarify that the DNA isolation was performed from pure fungal cultures and not from mixed environmental samples, thus avoiding host or non-target DNA contamination.

Comments 3: I recommend to extend the Discussion section and consider relevant information on the analogous genomic studies of other fungal pathogens.

Response 3: Thank you for your valuable comment. We appreciate this important suggestion. In response, we have extended the Discussion section to provide a broader context for our findings. Given the limited genomic research on N. laricinum isolates, we have incorporated comparisons with published genetic studies of other fungal pathogens (pages 11–12, lines 328–381).

Comments 4: The Conclusion section should be highlighted at the end of the manuscript.

Response 4: Thank you for your valuable comment. We appreciate your suggestion. In response, we have added a 5. Conclusion section at the end of the manuscript to highlight the main findings and implications of our study (page 12, lines 400–413).

Comments 5: Line 162 - "3. Genetic relationship among Chinese indigenous Pinewood Nematode". This raises questions, since the article is talking about a pathogenic fungus.

Response 5: Thank you for your insightful comment. We appreciate your careful reading. In response, we have corrected the section title on line 218 to: “3.3. Genetic relationship among Chinese Indigenous N. laricinum

Comments 6: Supplementary materials: The name of Supplementary file 1 is missing. Supplementary file 4. Please indicate that 23 N. laricinum isolates were studied (the word “isolates” is missing).

Response 6: Thank you for your insightful comments. We appreciate your careful review. In response, we have added the name “Supplementary File 1” on page 12, lines 416. Additionally, we have revised the title of Supplementary File 4 to include the word “isolates,” clarifying that the file pertains to the 23 N. laricinum isolates (page 12, lines 416–428).

Comments 7: The figures/tables/images/schemes are appropriate. They properly show the data. All illustrative materials are of satisfactory quality. However, to my opinion, Figures 4 and 5 are too small and will be difficult for a reader to perceive them.

Response 7: Thank you for your comments. We appreciate your careful review. In response, we have revised Figures 4 and 5 to improve readability by enlarging the font size and enhancing the clarity of the plots. Additionally, we have adjusted the layout to present the data more compactly and clearly for better visual interpretation.

Reviewer 2 Report

Larch is an economically important tree species in China and the larger geographic region. As such, important pathogens that affect its viability and productivity, such as Neofusicoccum laricinum, deserve to be investigated to identify the drivers of their pathogenicity and differences in pathogenicity among groups of isolates. The study design appears logical and the results give a good overview of the genetic variability and degree of differences and clustering among the isolates from different regions. However, a number of significant issues should be addressed before publication, especially concerning overview of existing knowledge and description of the applied methods. 

Regarding the overall study, the aims should be more precisely defined in the abstract and in the introduction section. The introduction section itself is very brief and does not address previously published literature in a sufficient manner. The methods are currently not adequately explained and in some aspects there are evident lapses or mistakes. For instance, sampling, culture isolation and DNA extraction methods are not discussed. DNA quality control is also not explained other than concentration measurement. The statement “Next-generation sequencing libraries were prepared using NovaSeq 6000 Reagent Kit (Illumina Inc., USA)“ appears incorrect as this is a sequencing kit and not a library preparation kit.  FastQC was used for quality control, but it is unclear which principles were followed during this control. The statement „Functional annotations of identified variants were performed by aligning them against multiple databases using NCBI BLAST v2.9.0“ should clearly state the databases and versions that were used for annotation. It is also insufficient to state that a specific R package was used, instead, the specific functions should also be specified. It is also confusing, why the sequencing data was not used to assemble de novo genome assemblies, as this would add value to the study. In the results section, the quality of the sequencing data should be described in more detail, by highlighting the quality (fraction >Q30, >Q20) and coverage of the reference genome. For the phylogenetic analysis, a tree with bootstrap support values should also be provided.

Author Response

Comments 1: Regarding the overall study, the aims should be more precisely defined in the abstract and in the introduction section. The introduction section itself is very brief and does not address previously published literature in a sufficient manner.

Response 1: Thank you for your comment. In response, we have revised both the abstract and the introduction to more clearly define the objectives of this study. Specifically, we added the following statement to the abstract: “This study aimed to investigate the genomic diversity, population structure, and potential adaptive mechanisms of N. laricinum across contrasting climatic regions” (page 1, lines 10–27). Additionally, we have substantially expanded the Introduction section to include more detailed background information and a comprehensive review of relevant genomic studies on Neofusicoccum species and other fungal pathogens. These revisions provide stronger context and rationale for our research approach (pages 1–2, lines 31–79).

Comments 2: The methods are currently not adequately explained and in some aspects there are evident lapses or mistakes. For instance, sampling, culture isolation and DNA extraction methods are not discussed.

Response 2: Thank you for your comment. In response, we have added a dedicated section (Section 2.2. N. laricinum isolation and DNA extraction) detailing the sampling strategy, culture isolation procedures using PDA medium, and DNA extraction using the SDS-based fungal DNA protocol. These revisions can improve methodological transparency and reproducibility (pages 3-4, lines 103-121).

Comments 3: DNA quality control is also not explained other than concentration measurement.

Response 3: Thank you for your comment. In response, we have updated “Section 2.3 Genomic DNA quality control, library preparation, and sequencing” to include a full description of the DNA quality control procedures. In addition to concentration measurements using Qubit, we assessed DNA integrity via 1% agarose gel electrophoresis and evaluated purity using OD260/OD280 ratios measured by a NanoDrop spectrophotometer. These details now show on page 4, lines 120–132.

Comments 4:The statement “Next-generation sequencing libraries were prepared using NovaSeq 6000 Reagent Kit (Illumina Inc., USA)“ appears incorrect as this is a sequencing kit and not a library preparation kit. FastQC was used for quality control, but it is unclear which principles were followed during this control.

Response 4: Thank you for your comment. We appreciate your careful reading. The original statement incorrectly referred to the NovaSeq 6000 Reagent Kit as a library preparation kit. We have corrected this by specifying that libraries were prepared using the ND627 kit (Vazyme Biotech Co., Ltd., Nanjing, China), while sequencing was performed using the DNBSEQ-T7 High-throughput Sequencing Set (MGI Tech Co., Ltd., Shenzhen, China). This clarification appears on section 2.3 Genomic DNA quality control, library preparation, and sequencing, page 4, lines 126–134.

We also have clarified that quality control was conducted using FastQC v0.11.8 to evaluate metrics such as per-base sequence quality, GC content, sequence duplication levels, and adapter contamination. Reads not meeting quality thresholds (Phred score < 20, length < 20 bp, or >10% N bases) were discarded before downstream analysis (page 4, lines 133–156).

Comments 5:The statement “Functional annotations of identified variants were performed by aligning them against multiple databases using NCBI BLAST v2.9.0” should clearly state the databases and versions that were used for annotation. It is also insufficient to state that a specific R package was used, instead, the specific functions should also be specified.

Response 5: Thank you for your comment. We appreciate your careful reading. In response, we have revised the manuscript to clearly specify the databases “eggNOG/COG [1], GO [2], KEGG [3], NCBI non-redundant protein database (Nr) [4], Pfam [5], and Swiss-Prot [6]” used for functional annotation, the thresholds applied, and the exact versions of all tools and functions (page 5, lines 167–176).

Comments 6: It is also confusing, why the sequencing data was not used to assemble de novo genome assemblies, as this would add value to the study.

Response 6: Thank you for your comment. We appreciate your suggestion regarding de novo genome assembly. While our current study focuses on variant calling relative to the reference genome due to its population-genomics scope, we agree that de novo assemblies could provide valuable insights. We have now added this as a recommended direction in the Discussion section for future work (page 11, lines 398–401).

Comments 7: In the results section, the quality of the sequencing data should be described in more detail, by highlighting the quality (fraction >Q30, >Q20) and coverage of the reference genome.

Response 7: Thank you for this valuable suggestion. We agree that these metrics are important for variant identification. We have added detailed information regarding sequencing data quality in the Results section (Section 3.2: Distribution of variants in Chinese indigenous N. laricinum), including Q20 and Q30 scores, total read numbers, and mapping coverage relative to the reference genome (Page 6, lines 202–214). Additionally, comprehensive quality metrics for each isolate have been provided in Supplementary File 1.

Comments 8: For the phylogenetic analysis, a tree with bootstrap support values should also be provided.

Response 8: Thank you for this valuable suggestion. We acknowledge the importance of bootstrap support in phylogenetic analyses for inferring evolutionary histories. However, given the absence of known ancestry sequences for N. laricinum, conducting robust evolutionary inference through bootstrap-supported trees is not feasible. Also, the primary purpose of the phylogenetic tree in our study is to visualize overall genetic relationships and population structure among N. laricinum isolates, rather than to reconstruct detailed evolutionary histories. Therefore, we applied a widely accepted approach in population genetics by constructing a Neighbor-Joining (NJ) tree based on identity-by-state (IBS) distance matrices using PLINK v2.0. This method is commonly used in genetic diversity and population structure studies and is not typically accompanied by bootstrap support values when used for exploratory or comparative purposes.

Examples of similar applications can be found in published studies such as:

Zhao, Z. et al. (2022). Population structure analysis to explore genetic diversity and geographical distribution characteristics of cultivated-type tea plant in Guizhou Plateau. BMC Plant Biology, 22, 55.

Kalinowski, S. (2009). How well do evolutionary trees describe genetic relationships among populations? Heredity, 102, 506–513.

We believe this approach is appropriate for our research goals and complements the findings from PCA and population structure analyses.